

# An efficient approach for treating composition-dependent diffusion within organic particles

Simon O'Meara[1], David O. Topping[1,2], Rahul A. Zaveri[3], and Gordon McFiggans[1]

[1]Centre for Atmospheric Science, School of Earth & Environmental Sciences, University of Manchester, Manchester, M13 9PL, UK
[2]National Centre for Atmospheric Science (NCAS), University of Manchester, Manchester, M13 9PL, UK
[3]Atmospheric Sciences and Global Change Division, Pacific Northwest National Laboratory, Richland, Washington 99352, USA

*Correspondence to:* Gordon McFiggans (g.mcfiggans@manchester.ac.uk)

**Abstract.**

Mounting evidence demonstrates that under certain conditions the rate of component partitioning between the gas- and particle-phase in atmospheric organic aerosol is limited by particle-phase diffusion. To date, however, particle-phase diffusion has not been incorporated to regional atmospheric models. An analytical rather than numerical solution to diffusion through
organic particulate matter is desirable because of its comparatively small computational expense in regional models. Current analytical models assume diffusion to be independent of composition, and therefore use a constant diffusion coefficient. To realistically model diffusion, however, it should be composition-dependent (e.g. due to the partitioning of components that plasticise, vitrify or solidify). This study assesses the modelling capability of an analytical solution to diffusion corrected to account for composition dependence against a numerical solution. Results show reasonable agreement when the gas-phase
saturation ratio of a partitioning component is constant and particle-phase diffusion limits partitioning rate ($< 10\%$ discrepancy in estimated radius change). However, when the saturation ratio of the partitioning component varies a generally applicable correction could not be found, indicating that existing methodologies are incapable of deriving a general solution. Until such time as a general solution is found, caution should be given to sensitivity studies that assume constant diffusivity. The correction was implemented in the polydisperse multi-process Model for Simulating Aerosol Interactions and Chemistry (MOSAIC),
and is used to illustrate how the evolution of number size distribution may be accelerated by condensation of a plasticising component onto viscous organic particles.

## 1   Introduction

The accurate simulation of atmospheric aerosol transformation has been identified as a key component of assessing aerosol impact on climate and health (Jacobson and Streets, 2009; Fiore et al., 2012; Boucher et al., 2013; Glotfelty et al., 2016).
However, comprehensive modelling of the physicochemical processes that determine aerosol transformation across large spatial and temporal scales can be challenging due to the limitations of computer power (Zaveri et al., 2008). While the majority



of processes in large-scale models are solved by numerical methods, analytical solutions offer less computational expense. Particle-phase diffusion may be solved both analytically, under certain assumptions, or numerically (Crank, 1975).

The advantage of an analytical solution over a numerical one is the decreased computer expense (e.g. Smith et al., 2003; Zobrist et al., 2011; Shiraiwa et al., 2012). The Euler forward step method of Zobrist et al. (2011) was observed to have

5 the shortest computer time of three published numerical methods for diffusion estimation (O'Meara et al., 2016). When a constant particle-phase diffusivity was assumed this method had a computer time approximately a factor of 20 greater than the analytical method presented in Zaveri et al. (2014) (with the numerical method using the minimum spatial resolution (20 shells) required for convergence of predicted equilibrium times, and the maximum change in component molecule number per time step recommended by Zobrist et al. (2011), while the analytical method used a conservative temporal resolution of $1 \times 10^3$ time

steps). To rigorously investigate the role of composition-dependent particle-phase diffusion in particulates containing organic components a multi-process large-scale model is required. An analytical-solution to particle-phase diffusion would make this much more practical than a numerical solution with respect to computer time.

Unlike gas-phase diffusion, which is already accounted for in regional-scale models by equations of gas/particle partitioning (Seinfeld and Pandis, 2006; Zaveri et al., 2008), particle-phase diffusion has not yet been included. Two outcomes of recent

studies, however, indicate that particle-phase diffusion may pose a limitation to mass transfer. The first is field and laboratory observations that indicate organic particulates existing in a glassy phase state (Zobrist et al., 2008; Virtanen et al., 2010; Vaden et al., 2011; Saukko et al., 2012). Second is the contribution of very low volatility organic compounds (Ehn et al., 2014; Tröstl et al., 2016) to particulate matter, since volatility and diffusivity show positive correlations (Kroll and Seinfeld, 2008; Koop et al., 2011).

Whether particle-phase diffusion exerts a significant influence on the transformation of organic particulate matter remains an unanswered question. A major advance was the incorporation of an analytical solution to composition-independent particle-phase diffusion into a growth equation for a spherical particle by Zaveri et al. (2014). In examples of constant particle-phase diffusion coefficients, it was shown that, with sufficiently low diffusivity, particle number size distributions could be greatly perturbed, though there was also a dependency on reaction rate and volatility. Using both analytical and numerical solutions to

mass transfer equations, Mai et al. (2015) also report particle-phase diffusion being limiting under certain conditions, with a dependency on accommodation coefficient, particle size, and volatility.

While the results of Zaveri et al. (2014) and Mai et al. (2015) are highly beneficial, they have not accounted for the possibility of composition-dependent diffusion (Vignes, 1966; Lienhard et al., 2014; Price et al., 2015; O'Meara et al., 2016). This is particularly relevant when considering the role of water, which is important because of its comparatively high abundance and

30 high self-diffusion coefficient (Starr et al., 1999; O'Meara et al., 2016). The potential for water exerting a plasticising effect on low diffusivity organic particles is particularly important because the constituent components are expected to be highly oxidised (Ehn et al., 2014; Tröstl et al., 2016) and therefore polar and likely water soluble (Zuend et al., 2008; Topping et al., 2013). While numerical solutions to composition-dependent diffusion are available (Zobrist et al., 2011; Shiraiwa et al., 2012; O'Meara et al., 2016), an analytical solution has not, to the author's knowledge, yet been published. Indeed, Zaveri et al. (2014)





state that the analytical solution requires incorporation of further complexity in the particle-phase: heterogeneously distributed reactant species, liquid-liquid phase separation and heterogenous (with regard to position) diffusivity.

How does radial heterogeneity of diffusivity arise? Atmospheric component concentrations and their partitioning coefficients will vary substantially in time and space (Donahue et al., 2006), leading to concentration gradients through particles. With sufficient difference in the self-diffusivity of the component to the diffusivity of the particle bulk initially (in the case of condensation) or at equilibrium state (in the case of evaporation), and sufficient abundance of the component in the vapour-phase (condensation) or particle-phase (evaporation), diffusion is likely to occur at a rate dependent on particle composition. An example would be a particle predominately composed of secondary organic material with a low diffusivity that was formed during a comparatively low relative humidity afternoon and present in the boundary layer. Relative humidity increases as evening progresses and air temperature decreases. The resulting condensation of water onto the outside of the particle establishes a concentration gradient, thereby inducing diffusion. The increased concentration of water will act to increase diffusivity near the surface, whilst diffusivity in the particle core remains low (Zobrist et al., 2011; Lienhard et al., 2014; Price et al., 2015; O'Meara et al., 2016).

The analytical solution is strictly valid under the following conditions: constant concentration of the diffusing component at the particle surface, constant particle size and constant diffusion coefficient (diffusivity). In deriving a correction for varying diffusion coefficient, therefore, corrections to variable surface concentration and particle size may be implicit, depending on the scenario. Thus in the results below, the derivation of a correction is first studied for the relatively simple case of a constant surface mole fraction (determined through equilibration with a constant gas-phase saturation ratio). Second, the case of variable surface mole fraction (due to equilibration with a variable gas-phase saturation ratio) is studied. In addition, the effects of composition-dependent diffusion on number size distribution are demonstrated.

## 2 Method

In the first part of the method the model setup will be described, including all assumptions made. A simple two component system was assumed, comprising one semi-volatile ($sv$) and one non-volatile component ($nv$) that were nonreactive. Both components were assigned a molecular weight of 100 g mol$^{-1}$ and a density of 1x10$^6$ g m$^{-3}$ (in the discussion it is shown that the model is sensitive to the ratio of the component molar volumes rather than absolute values of molecular weight or density). Ideality was assumed, therefore particle-phase volume was calculated by the addition of the product of each components' number of moles and molar volume. The initial particle-phase concentration was radially homogenous. For the purpose of deriving a solution to particle-phase diffusion independent of gas-phase diffusion the latter was assumed instantaneous. Therefore, in combination with the assumption of ideality, changes to the particle-phase surface mole fraction of the partitioning component implies equal changes to its gas-phase saturation ratio.





Fick's second law was solved by a numerical method; for a sphere, with spherical coordinates and with the diffusion coefficient ($D$) dependent on composition, this is (Crank, 1975):

$$\frac{\partial C_i(r,t)}{\partial t} = \frac{1}{r^2}\frac{\partial}{\partial r}\left(r^2 D_i(C_i)\frac{\partial C_i(r,t)}{\partial r}\right),\tag{1}$$

for component $i$, where $C$ is concentration, $r$ is radius and $t$ represents time. In this study $D$ followed a logarithmic depen-
5 dence on the mole fraction of the semi-volatile component :

$$D(x_{sv}) = (D^0_{sv})^{x_{sv}}(D^0_{nv})^{(1-x_{sv})},\tag{2}$$

where $D^0$ is the self-diffusion coefficient and $x$ is mole fraction. This equation fitted measurements reported in Vignes (1966) for ideal mixtures.

Equation 1 can be solved by several numerical methods (e.g. Zobrist et al., 2011; Shiraiwa et al., 2012), but here we use
the initial-boundary problem approach (Fi-PaD) as presented in O'Meara et al. (2016). This model operates by splitting the particle into concentric shells, each assumed to be homogeneously mixed. The shell representation allows the radial profile of concentration ($C$) and therefore diffusion coefficient ($D$) to be realised. Increased steepness of the $D$ gradient requires increased spatial resolution for accurate diffusion estimation. The volume of shells is revalued after every time step. Greater model temporal resolution is required with increased rates of volume change to account for the effect of particle size on
diffusion rate. Therefore, as described in O'Meara et al. (2016), a maximum radius change of 0.1% was allowed over a single time step, and the interval was iteratively shortened until this condition was met.

The analytical solution to diffusion is presented and described in Zaveri et al. (2014). For a non-reactive component with instantaneous gas-particle surface equilibration it is:

$$\frac{d\overline{C}_{a,i,m}}{dt} = 4\pi R^2_{\mathrm{p},m} N_m K_{p,i,m}(\overline{C}_{g,i} - \overline{C}_{a,i,m}S_{i,m}),\tag{3}$$

where $K_{p,i,m}$ is the overall mass transfer coefficient:

$$\frac{1}{K_{p,i,m}} = \frac{R_{p,m}}{5D_i}\left(\frac{C^*_{g,i}}{\sum_j \overline{C}_{a,j,m}}\right),\tag{4}$$

and $S_{i,m}$ is the saturation ratio:

$$S_{i,m} = \frac{C^*_{g,i}}{\sum_j \overline{C}_{a,j,m}},\tag{5}$$

where $_a$ and $_s$ represent the bulk and surface of the particle-phase, respectively, $_g$ represents the gas-phase, $j$ is the in-
25 dex for all components, $m$ is the index for size-bin, $R_{\mathrm{p}}$ is particle radius, $C^*_g$ is the effective saturation vapour concentration ($\mathrm{mol\,m^{-3}(air)}$), $\overline{C}$ represents the concentration in the bulk part of a phase and $N$ is the particle number concentration ($\mathrm{m^{-3}(air)}$).





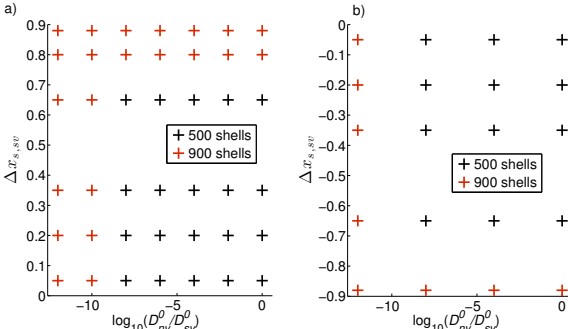

**Figure 1.** The shell resolution (given in the legend) distribution with $\Delta x_{s,sv}$ and $\log_{10}\left(\frac{D^0_{nv}}{D^0_{sv}}\right)$ used, for: a) +ve $\Delta x_{s,sv}$ and b) -ve $\Delta x_{s,sv}$.

The analytical solution treats the particle as a single body, i.e., it cannot resolve radial heterogeneity of concentration and therefore diffusion coefficient (the $D - r$ profile). In order for the diffusion coefficient in the analytical method to respond to composition variation therefore, $D$ was determined using eq. 2, which in turn used the bulk particle semi-volatile mole fraction ($x_{a,sv}$). Because $D$ and the correction factor (derivation described below) varied with composition, the analytical solution was

sensitive to temporal resolution. Analytical estimates were compared for a given scenario when the time steps of the Fi-PaD simulation were used and when a temporal resolution twice as fine was used. Results were identical, therefore the Fi-PaD resolution was considered sufficient for reliable analytical results.

Particles were assumed to initially have a radially homogenous concentration profile. Diffusion was then initiated by a change to the semi-volatile mole fraction at the particle surface ($\Delta x_{s,sv}$) to attain the equilibrium mole fraction $x_{s,sv,eq}$.

The radial heterogeneity of $D$ (in Fi-PaD) was therefore established through the setting of $D^0_{sv}$ and $D^0_{nv}$ and through the radial concentration gradient of the semi-volatile component resulting from diffusion. Since diffusion approaches equilibrium asymptotically, it is necessary to define an effective equilibrium point prior to complete equilibrium. We chose the $e$-folding state, which is when the absolute difference in component concentration at the surface and the bulk average (everything below the surface) decreases by a factor or $e$ from its initial value.

Fi-PaD estimates of the time required to reach the $e$-folding state (the $e$-folding time) converged as its spatial resolution increased (O'Meara et al., 2016). The spatial resolution required to attain a satisfactory degree of convergence increased with the gradient of the $D - r$ profile, which in turn was proportional to $\Delta x_{s,sv}$ and $D^0_{nv}/D^0_{sv}$. The maximum acceptable change for $e$-folding time following the addition of a further shell was set at 0.1 %. Based on this condition, fig. 1 shows the shell resolution used for combinations of $\Delta x_{s,sv}$ and $\log_{10}\left(D^0_{nv}/D^0_{sv}\right)$. The majority of scenarios used a conservative shell resolution, and only

where $|\Delta x_{s,sv}|$ and $|\log_{10}\left(D^0_{nv}/D^0_{sv}\right)|$ are both at a maximum for a given resolution was the convergence criteria neared.

As mentioned in the introduction, the correction of the analytical solution was for variation of not only the diffusion coefficient, but also particle size and surface concentration of the diffusing component. Consequently, corrections were derived and assessed for three scenarios of increasing complexity and generality. In the list of these scenarios below, the assumptions





of ideality and instantaneous gas-phase diffusion mean that the condition of the surface mole fraction of the semi-volatile component also represents that of its gas-phase saturation ratio:

    i) constant $x_{s,sv,eq}$, with initial/equilibrium $x_{s,sv} = 0$ for +ve $\Delta x_{s,sv}$/−ve $\Delta x_{s,sv}$

    ii) constant $x_{s,sv,eq}$, with initial/equilibrium $x_{s,sv} \neq 0$ for +ve $\Delta x_{s,sv}$/−ve $\Delta x_{s,sv}$

5    iii) variable $x_{s,sv,eq}$

For all scenarios the shell resolution distributions in fig. 1 were used to estimate the appropriate Fi-PaD spatial resolutions. $R_{\mathrm{p}} - t$ profiles estimated by the analytical solution were fit by eye to those of Fi-PaD to derive correction equations. $\Delta x_{s,sv}$ and $\log_{10}(D^0_{nv}/D^0_{sv})$ values across the ranges shown in fig. 1 were used, and the specific combinations shown in fig. 1 were used for the simplest derivation scenario (i) above). The analytical solution was found to have greater disagreement with the numerical solution for the condensation case than the evaporation case. Consequently fits were found for more combinations of $\Delta x_{s,sv}$ and $\log_{10}(D^0_{nv}/D^0_{sv})$ for the condensation case, as shown in fig. 1. An interpolation method was developed to estimate parameters for the correction equation between the values of $\Delta x_{s,sv}$ and $\log_{10}(D^0_{nv}/D^0_{sv})$ used for the equation derivation.

Finally, the following were incorporated into the Model for Simulating Aerosol Interactions and Chemistry (MOSAIC) (Zaveri et al., 2014): eq. 2, the correction equations and the interpolation method (eqs. 3-5 were already implemented). The temporal evolution of number size distributions was found for the case of condensation of a plasticiser and compared against an assumption of constant diffusivity. For elucidation of the effect on number size distribution of composition-dependent diffusion only the processes of gas/particle partitioning and particle-phase diffusion were modelled in MOSAIC.

## 3   Results

To begin, uncorrected analytical and Fi-PaD estimates of $e$-folding times were compared when $D$ was dependent on composition (eq. 2). Estimates were made for the $\Delta x_{s,sv}$ and $\log_{10}(D^0_{nv}/D^0_{sv})$ combinations in fig. 1, and the discrepancy is shown in fig. 2. For the case of +ve $\Delta x_{s,sv}$ (condensation) (fig. 2a), the analytical solution tends to underestimate diffusion rate, a result of being unable to resolve the plasticising effect of the semi-volatile component as it diffuses inward. Consequently, the discrepancy increases with increasing values of $|\Delta x_{s,sv}|$ and $|D^0_{nv}/D^0_{sv}|$, which together determine the plasticising effect. For -ve $\Delta x_{s,sv}$ (evaporation) (fig. 2b), this trend is reversed for comparatively high values of $|\Delta x_{s,sv}|$ and $|D^0_{nv}/D^0_{sv}|$ because the analytical solution is unable to resolve the solidifying effect of the non-volatile component as the semi-volatile component diffuses outward. The solidifying effect decreases with decreasing $|\Delta x_{s,sv}|$ and $|D^0_{nv}/D^0_{sv}|$, whereas the inaccuracy introduced to the analytical by changing particle size is unaffected by $|D^0_{nv}/D^0_{sv}|$, but increases with $|\Delta x_{s,sv}|$. The competing effects of these sources of inaccuracy produce the irregular contour layout at higher values of $|D^0_{nv}/D^0_{sv}|$.

Generally the analytical solution is much more accurate for -ve $\Delta x_{s,sv}$, reaching a maximum absolute disagreement around 0.6 orders of magnitude compared to 7.0 for +ve $\Delta x_{s,sv}$. This is attributed to the different characteristics of diffusion between the -ve and +ve $\Delta x_{s,sv}$ cases. In the former, diffusion in Fi-PaD is limited by $D$ near the particle surface, with a surface shell acting like a "crust". During early stages, the plasticising effect of the semi-volatile component on this "crust" leads to comparatively rapid diffusion out of the particle, but the strength of this effect decreases with concentration of the semi-





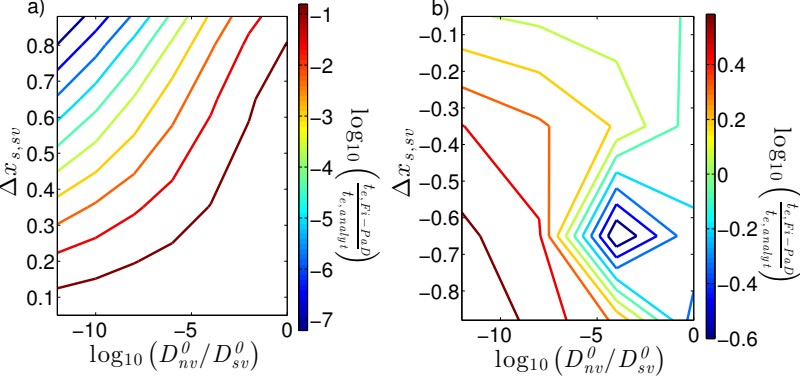

**Figure 2.** Ratio of Fi-PaD and analytical ($analyt$) estimated $e$-folding times ($t_e$) for: a) +ve $\Delta x_{s,sv}$ and b) -ve $\Delta x_{s,sv}$.

volatile component, so that the majority of the $e$-folding time is characterised by a gradual, relatively slow diffusion outward (see appendix for an example of the diffusion coefficient variation with radius for the evaporating case). The inability of the analytical solution to resolve the limiting diffusion near the surface leads to a greater rate of initial diffusion, however the consequent decrease in semi-volatile component concentration results in a $D$ value that replicates the slow diffusion phase of

Fi-PaD. In contrast, for +ve $\Delta x_{s,sv}$, diffusion is limited at the diffusion"front", which is the shell boundary between shells with the greatest radial gradient of concentration. Modelling movement of the "front" requires knowledge of the concentration gradient there, however the only information available to the analytical approach is the particle bulk concentration, leading to the large discrepancies seen.

To bring the analytical and numerical solutions into agreement, a correction factor is proposed for the analytical solution.

This will act on the diffusion coefficient to correct the diffusion rate (and is therefore denoted by $C_D$). Eq. 4 is thus modified to:

$$\frac{1}{K_{p,i,m}} = \frac{R_{p,m}}{5C_D D_i} \left( \frac{C_{g,i}^*}{\sum_j \overline{C}_{a,j,m}} \right). \tag{6}$$

To derive a function for $C_D$ first the simplest scenario of a single and instantaneous change in $x_{s,sv}$ with the initial/final $x_{s,sv} = 0$ for +ve $\Delta x_{s,sv}$/-ve $\Delta x_{s,sv}$ is investigated. The correction factor for $D$ required to bring analytical $R_p$ estimates into

agreement with those of Fi-PaD was found at each time step used by the latter model. The correction factor was then plotted against a metric for proximity to equilibrium; for +ve $\Delta x_{s,sv}$ this was the ratio of surface to bulk average $x_{sv}$, while for -ve $\Delta x_{s,sv}$, this was the absolute difference between surface and bulk average $x_{sv}$. This process was done for the model inputs shown in fig. 1 to determine whether a general equation form could be found that described the relationship between the $D$




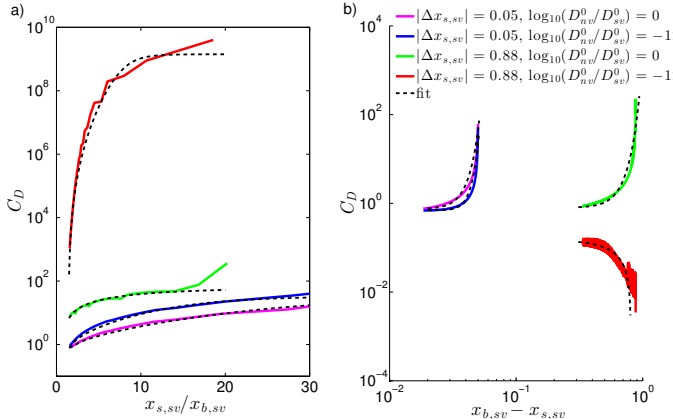

**Figure 3.** Examples of the correction factor for $D$ in the analytical solution ($C_D$) required to give agreement with radius estimates in Fi-PaD as a function of proximity to equilibrium (for which the metric depends on the sign of $\Delta x_{s,sv}$), for: a) +ve $\Delta x_{s,sv}$ and b) -ve $\Delta x_{s,sv}$. The model scenario is described in the legend, which applies to both plots. Fits are plotted using eqs. 7 and 8 for a) and b), respectively.

correction factor ($C_D$) and proximity to equilibrium. Examples are shown in fig. 3. The resulting general equations for +ve and -ve $\Delta x_{s,sv}$, respectively, are found to be:

$$C_D = (e^{(-(x_{s,sv}/x_{a,sv}-p_1)^{p_2})/p_3} + p_4)^{-1}, \tag{7}$$

and

5 $$C_D = e^{((x_{a,sv}-x_{s,sv})^{p_1})p_2} - p_3, \tag{8}$$

where $p_n$ is a parameter value, dependent on $\Delta x_{s,sv}$ (the change in semi-volatile surface mole fraction from the initial value (equal to the initial bulk particle mole fraction)) and $D^0_{nv}/D^0_{sv}$. Oscillations in $C_D$ occur for the case of $\Delta x_{s,sv} = -0.88$ and $\log_{10}(D^0_{nv}/D^0_{sv}) = -12$. This is attributed to the competing effects of changing particle size, which for a shrinking particle, acts to overestimate diffusion rate, and of a composition-dependent $D$, which for a solidifying particle acts to underestimate diffu-

10 sion time. As diffusion proceeds, slight variations in the relative strengths of these effects causes $C_D$ to oscillate. Nevertheless, an overall trend is discernible and can be described by eq. 8.

Parameter values for eqs. 7 and 8 were found through fitting by eye analytical $R_{\mathrm{p}} - t$ profiles with those of Fi-PaD for the model inputs shown in fig. 1 (values are provided in the appendix). To value the agreement between Fi-PaD and corrected analytical estimates, the following equation was used:

$$\% \quad \mathrm{error} = \left(\frac{R_{\mathrm{p},Fi-PaD,t} - R_{\mathrm{p},analyt,t}}{|R_{\mathrm{p},Fi-PaD,t=t_e} - R_{\mathrm{p},Fi-PaD,t=0}|}\right) 100, \tag{9}$$





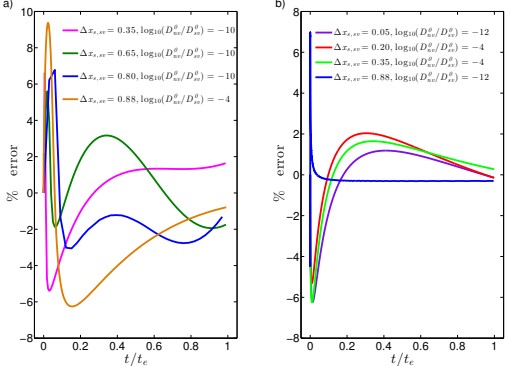

**Figure 4.** Examples of % error (eq. 9) of the analytical model with corrected $D$ when diffusion is composition-dependent (eq. 2), plotted against time normalised by the $e$-folding time. a) and b) are for +ve $\Delta x_{s,sv}$ and -ve $\Delta x_{s,sv}$, respectively, and scenarios are given in the legend.

where $analyt$ represents the corrected analytical model. Therefore, % error is the fraction of the total change in $R_{\mathrm{p}}$ comprised by the disagreement in model estimates of $R_{\mathrm{p}}$ at $t$.

For each marked $\Delta x_{s,sv}$ value in fig. 1, the marked $\log_{10}\left(D^{0}_{nv}/D^{0}_{sv}\right)$ scenario with greatest % error was identified. Of these scenarios, the four with greatest % error are shown in fig. 4 to demonstrate the cases of worst agreement. Fig. 4 shows that the

disagreement between analytical and Fi-PaD model estimates rarely exceeds $\pm6\%$, even for cases representing the extremes of model disagreement.

In order to have general applicability, such good agreement must be reproducible for intermediate values of $\Delta x_{s,sv}$ and $\log_{10}\left(D^{0}_{nv}/D^{0}_{sv}\right)$, i.e., when parameter values are interpolated between the points of fig. 1. Parameter relationships with $\Delta x_{s,sv}$ and $D^{0}_{nv}/D^{0}_{sv}$ varied substantially, requiring separate interpolation methods for each parameter. The interpolation methods are

presented in the appendix and were tested at $\Delta x_{s,sv}$ and $\log_{10}\left(D^{0}_{nv}/D^{0}_{sv}\right)$ comparatively far from those with known parameter values and spread across the variable space. Results are shown in fig. 5, again using the % error metric presented in eq. 9. They show that the low error produced for known parameter values is maintained when the interpolation method is applied.

Next, the case of a single and instantaneous change to $x_{s,sv}$ with the initial/final $x_{s,sv} \neq 0$ for +ve $\Delta x_{s,sv}$/−ve $\Delta x_{s,sv}$ is studied. For the +ve $\Delta x_{s,sv}$ case, the correction method described above was found to be transferable to any starting $x_{s,sv}$

through transformation of the $D$ dependence on $x_{sv}$. An effective self-diffusion coefficient of $nv$ ($D^{0}_{nv,eff}$) is set as the $D$ at the starting $x_{s,sv}$ (eq. 2), and the starting $x_{s,sv}$ for the analytical is set to 0. $D^{0}_{sv}$ is constant, but the equilibrium $x_{s,sv}$ ($x_{s,sv,eq}$) is changed to an effective value such that $D$ at equilibrium gives the same change in $D$ from the starting $x_{s,sv}$ as in the original scenario. Consistent with eq. 2 this effective $x_{s,sv,eq}$ is given by:

$$x_{s,sv,eq,eff} = \frac{\left(\log_{D^{0}_{nv,eff}}\left(\left(D^{0}_{nv}\right)^{(1-x_{s,sv,eq})}\left(D^{0}_{sv}\right)^{(x_{s,sv,eq})}\right) - 1\right)}{\left(\log_{D^{0}_{nv,eff}}\left(D^{0}_{sv}\right) - 1\right)},$$ (10)



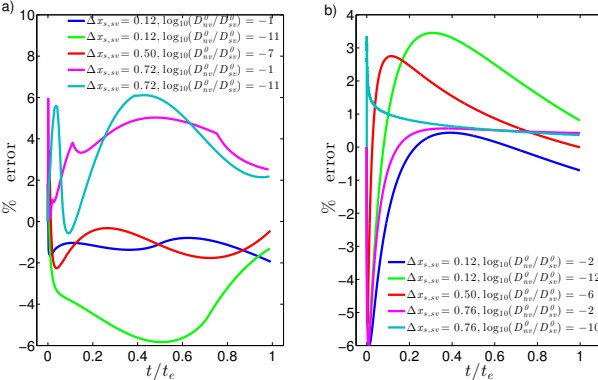

**Figure 5.** Examples of % error (eq. 9) of the analytical model with corrected $D$ and composition-dependent on diffusion (eq. 2), plotted against time normalised by the $e$-folding time. Parameter values for eqs. 7 and 8 were found through interpolation. a) and b) are for +ve $\Delta x_{s,sv}$ and -ve $\Delta x_{s,sv}$, respectively, and model setups are given in the legend.

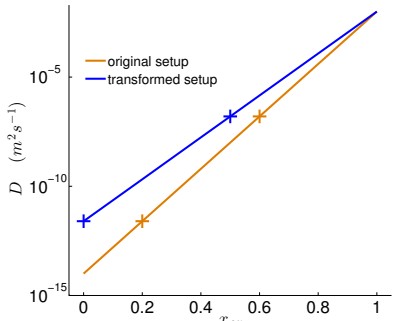

**Figure 6.** Example of the transformation of the $D$ dependence on $x_{sv}$ when the starting $x_{s,sv}$ (for condensation) or final $x_{s,sv}$ (for evaporation) $\neq 0$. In this example the original starting $x_{s,sv}$ was 0.2 and the equilibrium $x_{s,sv}$ was 0.6, while the original $D_{nv}^0$ was $1\times10^{-14}$ m$^2$s$^{-1}$ and $D_{sv}^0$ was $1\times10^{-2}$ m$^2$s$^{-1}$, as shown by the orange crosses. The effective starting and equilibrium $x_{s,sv}$ and effective $D_{nv}^0$ found by the transformation described in the main text are shown with blue crosses.

where $x_{s,sv,eq}$ and $D_{nv}^0$ are the original values. An example transformation to this effective model setup is shown in fig. 6. It can be seen that, compared to the original setup, $\Delta x_{s,sv}$ is increased. Although the transformed $D$ gradient with $x_{sv}$ is shallower than the original, therefore, this is offset in terms of diffusion rate by the increased radial gradient in $sv$ concentration.

A similar method can be applied to the evaporation scenario when the final $x_{s,sv} \neq 0$. $D_{nv,eff}^0$ is set equal to that at the final $x_{s,sv}$, and the final $x_{s,sv}$ is set to 0. Whereas for the +ve $\Delta x_{s,sv}$ case we found $x_{s,sv,eq,eff}$, now an effective start $x_{s,sv}$ ($x_{s,sv,0,eff}$) is required. The equation for this is the same as eq. 10, but with $x_{s,sv,eq,eff}$ replaced by $x_{s,sv,0,eff}$ and with $x_{s,sv,eq}$ replaced by $x_{s,sv,0}$. With regard to the transformed $D - x_{sv}$ profile (e.g. fig. 6), for a given pair of original start and finish $x_{s,sv}$ and a given pair of original self-diffusion coefficients, the transformation is the same for +ve and -ve $\Delta x_{s,sv}$.



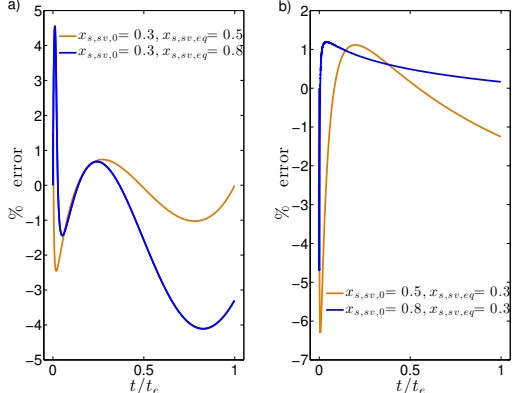

**Figure 7.** Agreement between corrected analytical and Fi-PaD estimates, using the metric given in eq. 9, for: a) +ve $\Delta x_{s,sv}$ and b) -ve $\Delta x_{s,sv}$, as shown in the legend. The start/finish $x_{s,sv} \neq 0$ for a)/b), therefore the transformation to an effective model setup (as described in the main text) was required. For both a) and b) $\log_{10}(D^0_{nv}/D^0_{sv}) = -12$.

To exemplify the deviation in analytical (with correction) estimates of diffusion rate from those of Fi-PaD when this transformation is applied, the cases of $\Delta x_{s,sv} = 0.2$ and $= 0.5$, and a comparatively large $\log_{10}(D^0_{nv}/D^0_{sv})$ of -12 were used. Estimates were compared using eq. 9. Results for +ve and -ve $\Delta x_{s,sv}$ are given in fig. 7, and demonstrate that the deviations are comparable to those when the transformation is not required (fig. 4).

Before moving onto a correction for the case of variable $x_{s,sv}$, the correction for constant $x_{s,sv}$ was implemented in MOSAIC to investigate the effect of composition-dependent diffusion on number size distribution. The same initial number size distribution as presented in Zaveri et al. (2014) (their fig. 11) was used. Reactions, coagulation, nucleation, emission and deposition were all turned off to gain the clearest demonstration of the diffusion effect. To maintain $x_{s,sv}$, the gas-phase concentration of the semi-volatile component was held constant and low particle-phase self-diffusion coefficients were used to
ensure that partitioning was not limited by diffusion in the vapour-phase. The model was run in Langrangian mode to prevent numerical error due to rebinning and resultant loss of information about the initial particle size.

To test the effect on the timescale of number size distribution change during condensation of a plasticising semi-volatile component, $\Delta x_{s,sv}$ was set to +0.88, from an initial particle-phase mole fraction of 0. The number size distribution following diffusion was found for $\log_{10}(D^0_{nv}/D^0_{sv})$ values of 0, -2 and -4, with $D^0_{nv}$ held constant at $1.0 \times 10^{-26}$ m$^2$s$^{-1}$. Simulations were
15 run until the largest particle had reached its $e$-folding state. The distributions after one tenth and at the end of the run time for the $\log_{10}(D^0_{nv}/D^0_{sv}) = -4$ case are shown in fig. 8a and fig. 8b, respectively, along with the initial distribution.

As expected, fig. 8 shows that the condensing component can significantly increase the rate of diffusion and therefore the rate at which the number size distribution evolves. For all values of $\log_{10}(D^0_{nv}/D^0_{sv})$ the form of the number size distribution shows the same characteristic of initially narrowing as smaller particles grow more quickly before widening again as these
20 particles equilibrate and larger particles grow. The degree of narrowing is similar between all cases, indicating that when a





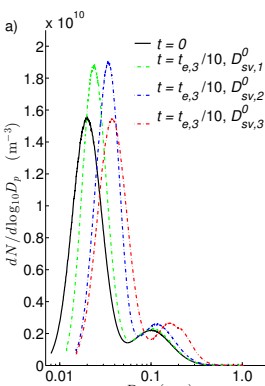 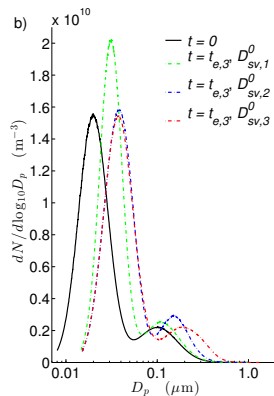

**Figure 8.** Number size distributions for $\log_{10}\left(D_{nv}^0/D_{sv}^0\right)$ = 0, -2 and -4, represented by $D_{sv,1}^0$, $D_{sv,2}^0$ and $D_{sv,3}^0$ respectively ($D_{nv}^0$ constant at $1.0 \times 10^{-26}$ m$^2$s$^{-1}$). $t_{e,3}$ is the time for the largest particle in the $\log_{10}\left(D_{nv}^0/D_{sv}^0\right)$ = -4 run to attain $e$-folding state. a) is the distribution at one tenth of $t_{e,3}$ and b) is that at $t_{e,3}$. $x_{s,sv}$ was increased instantaneously from 0.00 to 0.88 and then held constant.

plasticising effect occurs, the resulting acceleration of diffusion is similar across all particle sizes (consistent with the results of O'Meara et al. (2016)).

It is possible to set a constant diffusion coefficient in the analytical solution without correction that attains the same $e$-folding time as when the analytical solution with correction is used with a variable diffusion coefficient. For the case of

$D_{nv}^0 = 1.0 \times 10^{-26}$ and $D_{sv}^0 = 1.0 \times 10^{-22}$ m$^2$s$^{-1}$ and $\Delta x_{s,sv} = 0.88$ , the required constant diffusion coefficient was found to be $D_{con} = 4.4 \times 10^{-23}$ m$^2$s$^{-1}$. The % error (eq. 9) when the constant $D$ treatment is used is shown in fig. 9a. This figure shows that although the constant $D$ simulation does give the same $e$-fold time (agreement in radius estimate at $t/t_e = 1$), diffusion estimates about this point are different between the treatments of diffusion coefficient: beginning more quickly in the variable case before it slows relative to the constant case.

To test the effect of using a constant $D$ on a polydisperse population, this treatment is used to estimate number size distributions from MOSAIC and compared to estimates using the variable $D$ treatment. Using the same model setup as for fig. 8, the comparison is shown in fig. 9b-d . Results are shown for three times since run start as described in the figure. As expected from fig. 9a, if one focusses on the smaller particle sizes it can be seen that growth is initially quickest in the variable $D$ case (fig. 9b) but that growth in the constant $D$ case catches and exceeds that for variable $D$, leading to increased narrowing of the distribu-

tion (fig. 9d). Note that while this demonstration focuses on the smallest sizes, the same effect is true for all sizes, indeed in fig. 9d for $D_p$ around 0.1 $\mu$m it can be seen that particles are growing quicker in the variable treatment as diffusion initiates in these sizes. These discrepancies demonstrate the requirement for a correction to the analytical solution that is dependent on the proximity to equilibrium rather than a correction based on a constant $D$.

For the analytical solution to be generally applicable a correction when $x_{s,sv}$ varies prior to particle phase equilibration is

required. If the rate of $x_{s,sv}$ change is very low compared to particle-phase diffusion (particle-phase equilibration reached with negligible change of $x_{s,sv}$), or very high compared to particle-phase diffusion (no diffusion in the particle-phase before the





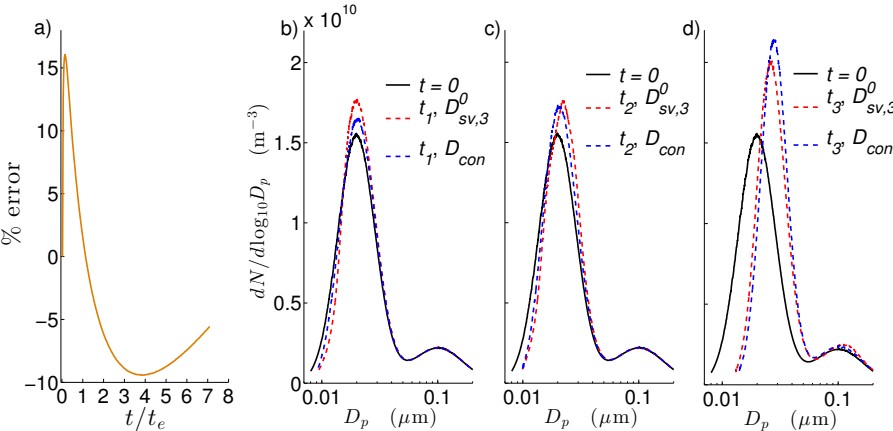

**Figure 9.** In a), the discrepancy (found using eq. 9) in estimated radius with model run time normalised to the $e$-folding time ($t_e$) when $x_{s,sv}$ is increased instantaneously from 0.00 to 0.88 for two diffusion coefficient treatments: i) $D_{nv}^0 = 1\text{x}10^{-26}$ and $D_{sv}^0 = 1\text{x}10^{-22}$ m$^2$s$^{-1}$ and ii) using the analytical without correction when $D$ is constant at $4.4\text{x}10^{-23}$ m$^2$s$^{-1}$. In later plots are the number size distributions for the same diffusion coefficient treatments, with red representing the former treatment (variable $D$) and blue the latter one (constant $D$)). In b) $t = 1.80\text{x}10^4$ s, c) $t = 4.50\text{x}10^4$ s and in d) $t = 5.76\text{x}10^5$ s since simulation start.

surface concentration reaches a constant value), no correction is needed. In between, however, a further correction dependent on the rate of $x_{s,sv}$ change is required. Changes to $x_{s,sv}$ may result from changes to the saturation ratio of the semi-volatile component. This may occur through a variety of ways, but in general is due to the sum of emission and production being different to that of deposition and destruction. Processes controlling gas-phase component concentrations occur at rates varying

by several orders of magnitude (e.g. reaction rate with OH radicals (Ziemann and Atkinson, 2012)). The rate of particle-phase diffusion may also vary by orders of magnitude, as it is dependent on the concentration and diffusivity of the diffusant as well as the diffusivity of the initial particle and the particle size (O'Meara et al., 2016).

Results shown to this point have been for a constant $x_{s,sv}$ (implying instantaneous particle surface-gas equilibration and a constant gas phase saturation ratio). Application of the corrections presented above (eqs. 7 and 8) to the variable case is not

straightforward as it is based on the difference between initial and equilibrium mole fractions and the particle is assumed to initially have a radially homogenous concentration profile. In the following passage is a description of a method to overcome this constraint for a given time profile of $x_{s,sv}$. This serves as a basis to explain the limits of this method to general application.

$x_{s,sv}$ was decreased from 0.88 to 0.00 with a sinusoidal profile, as shown in fig. 10a (curve p$_1$). The initial particle radius was $1\text{x}10^{-4}$ m, $D_{nv}^0 = 1\text{x}10^{-14}$ and $D_{sv}^0 = 1\text{x}10^{-10}$ m$^2$s$^{-1}$. The resulting $R_\text{p} - t$ profile using Fi-PaD is shown in fig. 10b.

For the analytical estimate to agree the correction equation is found to be:

$$C_D = e^{((p_4)^{p_1})p_2} - p_3,$$ (11)



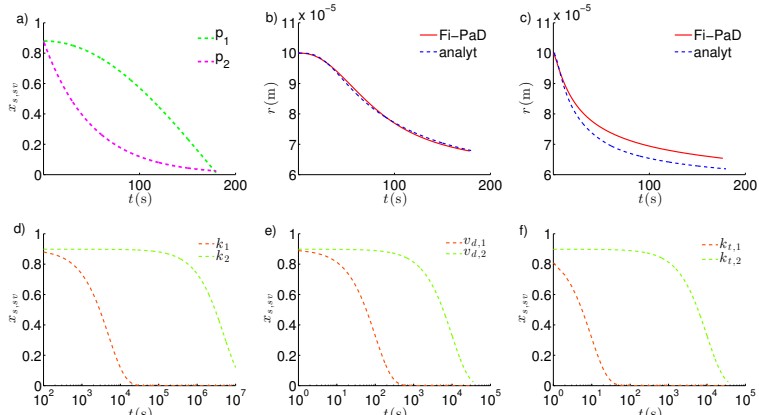

**Figure 10.** Plots demonstrating the limitation of the correction to cases of varying $x_{s,sv}$. In a) are the two temporal profiles of $x_{s,sv}$ used to test accuracy, while b) and c) show Fi-PaD and analytical (analyt) estimates of radius (the latter corrected using eqs. 11 and 12) for $x_{s,sv}$ temporal profiles $p_1$ and $p_2$, respectively. $D_{nv}^0 = 1 \times 10^{-14}$ and $D_{sv}^0 = 1 \times 10^{-10}$ m$^2$s$^{-1}$. In the lower row of plots are the range in rates of surface mole fraction change of a semi-volatile component assuming instantaneous equilibration with the gas-phase due to three processes: d) gas-phase chemical reaction with OH, with $k_1 = 1.0 \times 10^{-5}$ m$^3$ molecule$^{-1}$ s$^{-1}$ and $k_2 = 1.0 \times 10^{-8}$ m$^3$ molecule$^{-1}$ s$^{-1}$ (Ziemann and Atkinson, 2012); e) dry deposition to land surface, with $v_{d,1} = 1.0 \times 10^{-2}$ m s$^{-1}$ and $v_{d,2} = 1.0 \times 10^{-4}$ m s$^{-1}$ (Sehmel, 1980); f) condensation onto particles, with $k_{t,1} = 1.0 \times 10^{-1}$ s$^{-1}$ and $k_{t,2} = 1.0 \times 10^{-4}$ s$^{-1}$ (Sellegri et al., 2005; Whitehead et al., 2012).

where

$$p_4 = \frac{x_{a,sv}}{(\sin(\log_{10}(x_{rat,t_n} - x_{rat,t_{n-1}})/1.3 - 2.4)/4.6 + 1.1)}, \tag{12}$$

where $x_{rat}$ is the ratio of $x_{sv}$ in the particle bulk to that at the surface. The ratios at the start of the time step being solved for ($t_n$) and at the start of the previous time step ($t_{n-1}$) are used. $p_1$, $p_2$ and $p_3$ are the same as used for the original equation

(eq. 8) and $\Delta x_{s,sv}$ was set equal to the particle bulk $x_{sv}$.

This correction gives excellent agreement with the Fi-PaD estimate (fig. 10b). However, when used for a slightly different temporal profile of $x_{s,sv}$ (curve $p_2$ in fig. 10a), poorer agreement is attained. This indicates that the correction described in eqs. 11-12 is over fitted. This is unsurprising as it is dependent on the rate of change of the surface mole fraction of the semi-volatile component (through $x_{rat,tn} - x_{rat,tn-1}$). Consequently, we suggest that a generally applicable correction is only

possible with an a priori estimate of the rate of change of bulk to surface mole fraction ratio. However, the bulk mole fraction is the value being estimated, making a solution intractable using this methodology.

Also shown in fig. 10 is the expected range in rate of change of particle surface mole fraction of a semi-volatile component assumed to be in equilibrium with the gas-phase due to three processes: chemical reaction, dry deposition and condensation onto particles. The rates of change cover several orders of magnitude depending on the rate constants (given in the caption).

Comparing these rates to the $e$-folding times for particle phase diffusion given in O'Meara et al. (2016), it is clear that under certain scenarios the surface mole fraction change rate is similar to particle-phase diffusion rate. In this instance, the corrections





presented above break down. In contrast, when the particle-phase diffusion rate is much slower than the change in surface mole fraction of semi-volatile, a constant surface mole fraction may be assumed and the correction applied with the high accuracy presented above. This scenario is more likely to arise for particles with low diffusivity, and therefore of interest to particle-phase diffusion studies.

**4  Discussion**

As mentioned in the introduction, for a simple case of diffusion independent of composition, the computer time for the numerical solution is approximately 20 times as long as the analytical. However, this factor difference is expected to rise by 2-3 orders of magnitude for very steep gradients of diffusion coefficient with radius (O'Meara et al., 2016). Therefore, implementation of composition-dependent diffusion into a polydisperse multi-process aerosol model like MOSAIC through an analytical solution

is highly preferable to a numerical one. Here, equilibration between the gas- and particle-phase was assumed instantaneous, so that the surface mole fraction of the partitioning component was equal to its gas-phase saturation ratio.

For the limiting case of constant surface mole fraction of a semi-volatile component, here a correction to the analytical solution for when diffusivity is composition-dependent has been derived and validated against estimates from the numerical solution. A method to interpolate correction parameters between values of $\Delta x_{s,sv}$ (change to surface mole fraction that initiates

diffusion) and $D^0_{nv}/D^0_{sv}$ (ratio of component self-diffusion coefficients) was also derived and validated. A similar derivation was attempted for the case of variable surface mole fraction, however this was found to be of narrow applicability. This issue, along with the limitations of the correction for constant $x_{s,sv}$ are discussed below.

In favour of the correction is its independence of particle size. In both solutions (numerical and analytical), diffusion rates have a square dependence on particle size, therefore the ratios of estimated diffusion rate are constant across sizes (all else being

equal), as is the correction. Similarly, the correction is independent of absolute values of $D^0_{nv}$ and $D^0_{sv}$ and only dependent on the ratio of component self-diffusion coefficients: $\log_{10}(D^0_{nv}/D^0_{sv})$.

Although the correction is applicable across particle sizes and values of $D^0_{nv}$ and $D^0_{sv}$, it is specific to the ratio of component molar volumes used here, which is 1:1. The change in particle size due to partitioning depends on the molar volumes of components. The response of diffusion rate to a change in molar volume is different between the models and is non-linear in

each. For quantifying model sensitivity to molar volume, a further complication is the variation of diffusivity with both molar mass and density (Koop et al., 2011).

To gain an indication of the model disagreement arising from changing molar volume when the corrected analytical model is used, expected ranges of molar mass (M) and density ($\rho$) for atmospheric organic components were found. Barley et al. (2011) show that M is likely to be in the range $1\text{x}10^2$ to $3\text{x}10^2$ g mol$^{-1}$ and Topping et al. (2011) demonstrate that $\rho$ is likely to

30 be between $1.2\text{x}10^6$ to $1.6\text{x}10^6$ g m$^{-3}$. The maximum expected molar volume for the semi-volatile component was therefore given by using M = $3\text{x}10^2$ g mol$^{-1}$ and $\rho$ = $1.2\text{x}10^6$ g m$^{-3}$. A relatively large effect from the changed molar volume was gained through using $\Delta x_{s,sv} = \pm 0.88$. Furthermore, the proportion of the correction attributed to particle size change rather than $D$ composition dependence, is greatest for $\log_{10}(D^0_{nv}/D^0_{sv}) = 0$, therefore this was used to maximise the effect of varying





molar volume on model agreement. For the +ve and -ve $\Delta x_{s,sv}$ cases, the maximum observed % error (eq. 9) was -58.0 and 29.0 %, respectively. Given this large discrepancy and the complexity of the model responses, we recommend further work to investigate correction dependence on molar volume.

A further limitation of the presented correction is its specificity to the $D$ dependence on composition. Here we have assumed a logarithmic dependence on $x_{sv}$, however, measurements have reported sigmoidal and irregular dependencies resulting from changes to phase state and/or non-ideality (e.g. Vignes, 1966; Lienhard et al., 2014; Price et al., 2015). An indication of model disagreement generated by varying the $D$ dependence was found by calculating the % error for several dependencies; all were based on a sigmoidal function, however, the steepness at the "cliff-edge" was varied, as shown in fig. 11a. Also shown here is the logarithmic dependence used to find the presented correction. A $\log_{10}(D_{nv}^0/D_{sv}^0) = -12$ and $\Delta x_{s,sv} = \pm 0.88$ were used because these provide the most stringent test of estimation capability. The dependencies were used in both the Fi-PaD and analytical model, with the latter using the correction method for the logarithmic dependence. The resulting discrepancies in estimated particle radius are shown in figs. 11b and 11c.

Fig. 11b shows that for +ve $\Delta x_{s,sv}$, the analytical method increasingly overestimates initial diffusion with increasing sigmoidal function steepness, indicating the correction is too great when the ratio of surface to bulk $x_{sv}$ is high. The reason is that, with the dependencies used, increased steepness causes increased resistance to inward semi-volatile diffusion at low $x_{sv}$. As surface to bulk $x_{sv}$ ratio decreases in the analytical, so does the correction factor (fig. 3a), and Fi-PaD estimates begin to converge on the analytical. For the least steep sigmoidal dependence, diffusion in Fi-PaD overtakes the corrected analytical around 0.3 $t/t_e$. This occurs after some initial diffusion and is therefore attributed to Fi-PaD diffusion occurring quickly relative to the logarithmic dependence once the bulk concentration of the semi-volatile has been raised. This is demonstrated in fig. 11a, where for the least steep sigmoidal dependence, above $x_{sv} \approx 0.3$ the same change in $x_{sv}$ gives a greater increase in diffusivity than in the logarithmic dependence.

Results for -ve $\Delta x_{s,sv}$ are shown in fig. 11c, which shows that the analytical solution initially underestimates diffusion. This is attributed to the increasing plasticising effect of the semi-volatile on the surface crust of the particle with increasing steepness of the sigmoidal "cliff-edge". Once $x_{sv}$ has decreased however, the analytical shows a tendency to overestimate diffusion. The plasticising effect can quickly decrease (fig. 11a), and the surface crust imposes a greater impediment to diffusion. The correction factor (which acts to decelerate diffusion (fig. 3b)) found from the logarithmic dependence is insufficient to replicate this for the steepest dependency.

As fig. 11 shows, the presented correction is limited in its generality with regards to diffusion coefficient dependence on composition. Along with the effect of molar volume on diffusion, however, it is conceivable that this could be overcome through a more advanced correction similar in approach to that presented. In contrast, results indicate that improving the accuracy of the correction for the case of changing particle surface mole fraction is not attainable, since this requires a priori knowledge of the particle-phase diffusion rate (the value being estimated). Nevertheless, for studies into particle-phase diffusion limitation on particle transformation, it is possible that the surface mole fraction will vary quickly compared to particle-phase diffusion, allowing the assumption of a constant surface mole fraction and therefore accurate application of the correction presented here.





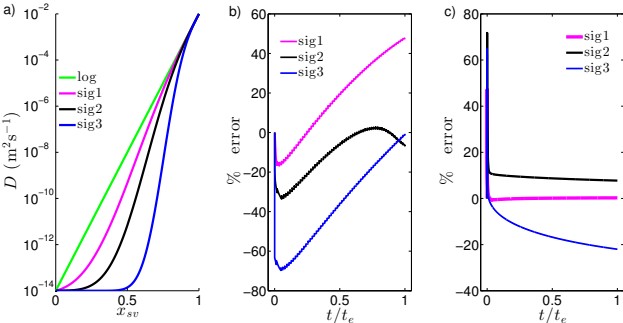

**Figure 11.** Plot a) shows the logarithmic dependency of diffusion coefficient on mole fraction on which the presented correction is derived and the sigmoidal dependencies for which it was tested. In b) and c) is the analytical model error (eq. 9) when the sigmoidal dependencies given in a) were used. b) +ve $\Delta x_{s,sv}$ and c) -ve $\Delta x_{s,sv}$, and for both plots $|\Delta x_{s,sv}| = 0.88$ and $\log_{10}(D_{nv}^0/D_{sv}^0) = -12$.

Without a general analytical solution (e.g. allowing for varying surface mole fraction), thorough evaluation of particle-phase diffusion influence on particulate transformation remains limited. The correction for constant surface mole fraction of the semi-volatile component, however, offers improved computer efficiency (compared to numerical methods) of evaluating particle-phase diffusion effects, such as in Berkemeier et al. (2013) and Mai et al. (2015). It may also be of use for the inference

of diffusivity from laboratory studies, if the rates of semi-volatile gas-phase saturation ratio change and gas-phase diffusion are much greater than the particle-phase diffusion rate (Zobrist et al., 2011; Lienhard et al., 2014; Steimer et al., 2015).

## 5 Conclusions

For accurate simulation of the transformation of particulates containing organic components, the analytical solution to diffusion must account for composition-dependent diffusion rate. To do this, a correction to the analytical solution was investigated

based on estimates from the numerical solution of the partial differential equation for diffusion. A correction was derived for the limiting case of a constant surface mole fraction of the diffusing component (equal to a constant gas-phase saturation ratio when assuming equilibration between the gas- and particle-phase). The corrected analytical solution shows good agreement with the numerical one, rarely exceeding 8 % deviation in estimated particle radius change.

The verified correction is currently limited to conditions of similar molar volume between the partitioning component and

15 the particle average, and of a logarithmic dependence of diffusion coefficient on partitioning component mole fraction. These limitations may be overcome through an advanced correction. However, a correction for the more general case of variable surface mole fraction of the diffusing component (e.g., due to varying gas-phase saturation ratio) was found to depend on the rate of change of the ratio of bulk to surface mole fraction. A correction based on the analytical approach presented here is therefore not viable because it requires a priori knowledge of the value to be estimated: the particle bulk mole fraction. A

20 different approach to modifying the analytical solution to diffusion is thus required to make it generally applicable.





   To determine whether an expression for particle-phase diffusion is required in a regional model, an evaluation of the sensitivity of organic particle properties to diffusion is desirable. This study builds on previous investigations toward allowing such a sensitivity analysis, and enables it for the limiting case of particles with sufficiently low diffusivity that changes to the particle surface mole fraction of the partitioning component occur much more quickly than particle-phase diffusion. Work remains, however, to create a generally applicable realistic and efficient diffusion model for particulates containing organic components. Until this is achieved, studies of aerosol kinetic regimes conducted under limiting scenarios such as diffusion independent of composition, should be interpreted cautiously because of their limited applicability to the real atmosphere. In particular, the comparatively high abundance and high self-diffusion coefficient of water means that its role in plasticising or vitrifying particles through condensation and evaporation, respectively, must be accounted for when assessing the effect of particle-phase diffusion on partitioning.



## Nomenclature

| | |
|---|---|
| $a$ | particle-phase bulk |
| $con$ | denotes a constant value |
| $C$ | concentration $(\mathrm{mol\,m^{-3}})$ |
| $\overline{C}$ | concentration in bulk part of a phase $(\mathrm{mol\,m^{-3}})$ |
| $C*$ | effective saturation vapour concentration $(\mathrm{mol\,m^{-3}\,(air)})$ |
| $C_D$ | diffusion coefficient correction |
| $D$ | diffusion coefficient $(\mathrm{m^2\,s^{-1}})$ |
| $eff$ | denotes an effective value |
| $eq$ | equilibrium state |
| $\mathrm{Fi-PaD}$ | Fick's Second Law solved by partial differential equation |
| $g$ | gas-phase |
| $i$ | a component |
| $j$ | all components |
| $k_n$ | chemical reaction rate $(\mathrm{m^3\,molecule^{-1}\,s^{-1}})$ |
| $k_t$ | condensation sink rate $(\mathrm{s^{-1}})$ |
| $K$ | mass transfer coefficient $(\mathrm{m\,s^{-1}})$ |
| $m$ | index for size-bin |
| $M$ | molar mass $(\mathrm{g\,mol^{-1}})$ |
| $\mathrm{MOSAIC}$ | Model for Simulating Aerosol Interactions and Chemistry |
| $N$ | particle number concentration $(\mathrm{m^{-3}\,air})$ |
| $nv$ | non-volatile component |
| $\rho$ | density $(\mathrm{g\,m^{-3}})$ |
| $p_n$ | correction equation parameter |
| $p$ | subscript denotes particle-phase |
| $r$ | radius $(\mathrm{m})$ |
| $rat$ | denotes a ratio |
| $R_p$ | total particle radius $(\mathrm{m})$ |
| $s$ | particle-phase surface |
| $sv$ | semi-volatile component |
| $t$ | time $(\mathrm{s})$ |
| $t_e$ | $e$-folding time $(\mathrm{s})$ |
| $t_n$ | a time after $n$ number of time steps $(\mathrm{s})$ |
| $v_d$ | deposition velocity $(\mathrm{m\,s^{-1}})$ |
| $x$ | mole fraction |





# Appendix A

| $\log_{10}(D^0_{nv}/D^0_{sv})$ | 0 | -2 | -4 | -6 | -8 | -10 | -12 |
|---|---|---|---|---|---|---|---|
| $\Delta x_{s,sv}$ | | | | $p_1$ | | | |
| 0.05 | 2.3000 | 2.1500 | 2.0000 | 1.9500 | 1.9100 | 1.8800 | 1.8800 |
| 0.20 | 1.8500 | 1.7800 | 1.7000 | 1.6400 | 1.6200 | 1.6000 | 1.5700 |
| 0.35 | 1.6000 | 1.5800 | 1.5700 | 1.5600 | 1.5600 | 1.5200 | 1.5200 |
| 0.65 | 1.4300 | 1.4500 | 1.4500 | 1.4700 | 1.4600 | 1.4500 | 1.4600 |
| 0.80 | 1.0696 | 1.3400 | 1.4300 | 1.4397 | 1.4348 | 1.4400 | 1.4550 |
| 0.88 | 0.0000 | 0.9300 | 1.0100 | 1.4238 | 1.4215 | 1.4400 | 1.4800 |
| | | | | $p_2$ | | | |
| 0.05 | $7.1430 \times 10^{-1}$ | $6.2500 \times 10^{-1}$ | $6.2500 \times 10^{-1}$ | $5.9170 \times 10^{-1}$ | $5.6820 \times 10^{-1}$ | $5.4050 \times 10^{-1}$ | $5.1280 \times 10^{-1}$ |
| 0.20 | $6.5070 \times 10^{-1}$ | $5.1900 \times 10^{-1}$ | $4.8950 \times 10^{-1}$ | $4.7990 \times 10^{-1}$ | $4.5000 \times 10^{-1}$ | $4.1930 \times 10^{-1}$ | $4.1670 \times 10^{-1}$ |
| 0.35 | $6.2500 \times 10^{-1}$ | $4.762 \times 10^{-1}$ | $4.348 \times 10^{-1}$ | $4.348 \times 10^{-1}$ | $3.7040 \times 10^{-1}$ | $3.7040 \times 10^{-1}$ | $3.7040 \times 10^{-1}$ |
| 0.65 | $4.2550 \times 10^{-1}$ | $3.9220 \times 10^{-1}$ | $3.7040 \times 10^{-1}$ | $3.4480 \times 10^{-1}$ | $3.3333 \times 10^{-1}$ | $3.1250 \times 10^{-1}$ | $3.1250 \times 10^{-1}$ |
| 0.80 | $2.6810 \times 10^{-1}$ | $3.0660 \times 10^{-1}$ | $2.9960 \times 10^{-1}$ | $2.9150 \times 10^{-1}$ | $2.8710 \times 10^{-1}$ | $2.8500 \times 10^{-1}$ | $2.8340 \times 10^{-1}$ |
| 0.88 | $3.0000 \times 10^{-1}$ | $4.8000 \times 10^{-1}$ | $6.0000 \times 10^{-1}$ | $2.9150 \times 10^{-1}$ | $2.8710 \times 10^{-1}$ | $2.6000 \times 10^{-1}$ | $2.7000 \times 10^{-1}$ |
| | | | | $p_3$ | | | |
| 0.05 | 3.0000 | 2.1500 | 2.0000 | 1.8000 | 1.6000 | 1.3600 | 1.2200 |
| 0.20 | 2.5346 | 1.3800 | 1.0000 | $7.4000 \times 10^{-1}$ | $6.2000 \times 10^{-1}$ | $4.9810 \times 10^{-1}$ | $4.1670 \times 10^{-1}$ |
| 0.35 | 2.1000 | 1.0000 | $6.2500 \times 10^{-1}$ | $4.4050 \times 10^{-1}$ | $3.3560 \times 10^{-1}$ | $2.7400 \times 10^{-1}$ | $2.2222 \times 10^{-1}$ |
| 0.65 | 1.1000 | $5.2630 \times 10^{-1}$ | $3.1150 \times 10^{-1}$ | $2.7140 \times 10^{-1}$ | $1.6450 \times 10^{-1}$ | $1.3230 \times 10^{-1}$ | $1.1170 \times 10^{-1}$ |
| 0.80 | $5.6500 \times 10^{-1}$ | $3.4500 \times 10^{-1}$ | $2.1700 \times 10^{-1}$ | $1.5700 \times 10^{-1}$ | $1.2200 \times 10^{-1}$ | $1.0100 \times 10^{-1}$ | $9.0500 \times 10^{-2}$ |
| 0.88 | $8.0000 \times 10^{-1}$ | $4.1000 \times 10^{-1}$ | $2.4000 \times 10^{-1}$ | $1.2950 \times 10^{-1}$ | $1.0300 \times 10^{-1}$ | $9.0500 \times 10^{-2}$ | $8.2000 \times 10^{-2}$ |
| | | | | $p_4$ | | | |
| 0.05 | $3.0000 \times 10^{-2}$ | $3.0000 \times 10^{-2}$ | $3.0000 \times 10^{-2}$ | $3.0000 \times 10^{-2}$ | $3.0000 \times 10^{-2}$ | $3.0000 \times 10^{-2}$ | $3.0000 \times 10^{-2}$ |
| 0.20 | $2.5700 \times 10^{-2}$ | $1.9600 \times 10^{-2}$ | $1.1500 \times 10^{-2}$ | $1.5900 \times 10^{-2}$ | $3.2000 \times 10^{-3}$ | $1.4000 \times 10^{-3}$ | $1.5000 \times 10^{-3}$ |
| 0.35 | $2.2000 \times 10^{-2}$ | $1.0000 \times 10^{-2}$ | $5.0000 \times 10^{-3}$ | $3.5000 \times 10^{-3}$ | $4.0000 \times 10^{-4}$ | $1.0000 \times 10^{-4}$ | $9.0000 \times 10^{-5}$ |
| 0.65 | $1.700 \times 10^{-2}$ | $2.0000 \times 10^{-3}$ | $6.0000 \times 10^{-4}$ | $4.0000 \times 10^{-5}$ | $6.0000 \times 10^{-6}$ | $7.0000 \times 10^{-7}$ | $1.2000 \times 10^{-7}$ |
| 0.80 | $1.5600 \times 10^{-2}$ | $9.8400 \times 10^{-4}$ | $1.1000 \times 10^{-4}$ | $3.8000 \times 10^{-6}$ | $5.2700 \times 10^{-7}$ | $3.8000 \times 10^{-8}$ | $4.5000 \times 10^{-9}$ |
| 0.88 | $3.0000 \times 10^{-2}$ | $7.3100 \times 10^{-4}$ | $2.0000 \times 10^{-4}$ | $1.2100 \times 10^{-6}$ | $1.2200 \times 10^{-7}$ | $6.9900 \times 10^{-9}$ | $7.0700 \times 10^{-10}$ |

**Table A1.** Eq. 7 parameter values found for +ve $\Delta x_{s,sv}$.

| $\log_{10}(D^0_{nv}/D^0_{sv})$ | 0 | -4 | -8 | -12 |
|---|---|---|---|---|
| $\Delta x_{s,sv}$ | | $p_1$ | | |
| 0.05 | 2.8100 | 2.8600 | 2.9200 | 3.0000 |
| 0.20 | 3.2300 | 3.5300 | 3.4600 | 2.0000 |
| 0.35 | 3.6500 | 4.4000 | 4.0000 | 2.0000 |
| 0.65 | 5.0000 | 8.0000 | 5.0000 | 2.0000 |
| 0.88 | 6.0000 | $1.1000 \times 10^1$ | 7.0000 | 1.9000 |
| | | $p_2$ | | |
| 0.05 | $8.0000 \times 10^3$ | $8.0000 \times 10^3$ | $8.0000 \times 10^3$ | $8.0000 \times 10^3$ |
| 0.20 | $3.5000 \times 10^2$ | $3.0000 \times 10^2$ | $1.0000 \times 10^2$ | -1.6000 |
| 0.35 | $1.0000 \times 10^2$ | $5.0000 \times 10^1$ | -1.0000 | -1.6000 |
| 0.65 | $2.3000 \times 10^1$ | $1.2000 \times 10^1$ | -1.0000 | $-4.0000 \times 10^{-1}$ |
| 0.88 | 7.0000 | 3.0000 | $5.6000 \times 10^{-1}$ | $-2.0000 \times 10^{-1}$ |
| | | $p_3$ | | |
| 0.05 | $4.0000 \times 10^{-1}$ | $4.2000 \times 10^{-1}$ | $4.0000 \times 10^{-1}$ | $4.2000 \times 10^{-1}$ |
| 0.20 | $3.2000 \times 10^{-1}$ | $4.1000 \times 10^{-1}$ | $5.0000 \times 10^{-1}$ | $5.2000 \times 10^{-1}$ |
| 0.35 | $2.5000 \times 10^{-1}$ | $4.0000 \times 10^{-1}$ | $5.8000 \times 10^{-1}$ | $6.2000 \times 10^{-1}$ |
| 0.65 | 0.0000 | $5.0000 \times 10^{-1}$ | $6.7000 \times 10^{-1}$ | $7.6000 \times 10^{-1}$ |
| 0.88 | $-1.0000 \times 10^{-1}$ | $5.8000 \times 10^{-1}$ | $7.8000 \times 10^{-1}$ | $8.5000 \times 10^{-1}$ |

**Table A2.** Eq. 8 parameter values found for -ve $\Delta x_{s,sv}$.





| $\log_{10}\left(D^0_{nv}/D^0_{sv}\right)$ | 0 | -2 | -4 | -6 | -8 | -10 | -12 |
|---|---|---|---|---|---|---|---|
| $p_i$ | | | | Method Code | | | |
| $p_1$ | 10L | 10L | 10S | 10S | 10S | 10S | 10S |
| $p_2$ | 10S | 10S | 01L | 01L | 01L | 01L | 01L |
| $p_3$ | 11L | 11L | 11S | 11S | 11S | 11S | 11S |
| $p_4$ | 10S | 10S | 10S | 10S | 10S | 10S | 10S |
| $\Delta x_{s,sv}$ | < 0.12 | ≥ 0.12 < 0.20 | ≥ 0.20 < 0.50 | ≥ 0.50 < 0.80 | ≥ 0.80 | | |
| $p_i$ | | | | Method Code | | | |
| $p_1$ | 01S | 01S | 01L | 01L | 01L | | |
| $p_2$ | 01S | 01S | 01L | 01S | 01L | | |
| $p_3$ | 01S | 01S | 11L | 01S | 01S | | |
| $p_4$ | 01S | 01S | 01L | 11L | 11L | | |

**Table A3.** Interpolation method for parameters in Eq. 7 (for +ve $\Delta x_{s,sv}$). Interpolation is done with respect to $\Delta x_{s,sv}$ and $\log_{10}\left(D^0_{nv}/D^0_{sv}\right)$ separately; the method for the former is given in the upper part of the table and for the latter see the lower part. The first number in each code represents whether the $\log_{10}$ of parameter values was taken (1 for yes, 0 for no), the second number indicates whether the $\log_{10}$ of the variable was taken (1 for yes, 0 for no), the final letter represents the form of the interpolation: L and S for linear and spline, respectively.

| $\log_{10}\left(D^0_{nv}/D^0_{sv}\right)$ | 0 | -4 | -8 | -12 |
|---|---|---|---|---|
| $p_i$ | | | Method Code | |
| $p_1$ | 00L | 00L | 00L | 00L |
| $p_2$ | 11L | 11L | 1(2)1L | 1(2)0L |
| $p_3$ | 00L | 00L | 00L | 00L |
| $\Delta x_{s,sv}$ | < 0.27 | ≥ 0.27 < 0.65 | ≥ 0.65 | |
| $p_i$ | | | Method Code | |
| $p_1$ | 01L | 01L | 01L | |
| $p_2$ | 1(2)1S ($D_r < -8$) | 1(2)1S ($D_r < -4$) | 1(1.1)1S ($D_r \geq -6$, $D_r \leq -4$) | |
| | 01L ($D_r \geq -8$) | 01L ($D_r \geq -4$) | 1(2)1L ($D_r > -4$, $D_r < -6$) | |
| $p_3$ | 1(2)1S | 1(2)1S | 1(2)1S | |

**Table A4.** Interpolation method for parameters in Eq. 8 (for -ve $\Delta x_{s,sv}$). Interpolation is done with respect to $\Delta x_{s,sv}$ and $\log_{10}\left(D^0_{nv}/D^0_{sv}\right)$ separately; the method for the former is given in the upper part of the table and for the latter see the lower part. The first number in each code represents whether the $\log_{10}$ of parameter values was taken (1 for yes, 0 for no). Because parameters are sometimes negative, to gain a real result from the logarithm, a constant must be added to the parameters, if this is the case this constant is given in brackets beside the first code number (note that once interpolation is complete this constant is subtracted from the result). The second number indicates whether the $\log_{10}$ of the variable was taken (1 for yes, 0 for no), the final element represents the form of the interpolation: L and S for linear and spline, respectively. For $p_2$, when interpolating with respect to $\log_{10}\left(D^0_{nv}/D^0_{sv}\right)$, the interpolation method depends on the value of this variable, which is denoted $D_r$.





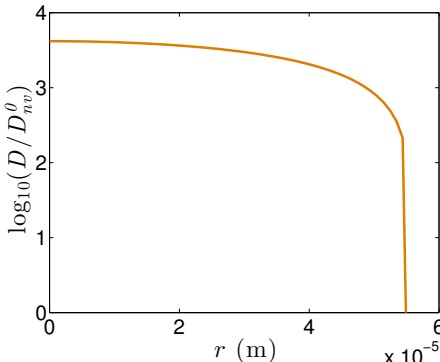

**Figure A1.** The logarithm of the ratio of the diffusion coefficient throughout an example particle to the self-diffusion coefficient of the non-volatile component, from the particle centre (at 0 m) to its surface. In this example, $\log_{10}(D_{nv}^{0}/D_{sv}^{0})$ =-12, and $x_{s,sv,eq} = 0$, and initial $x_{s,sv} = 0.88$.

*Author contributions.* S. O'Meara completed the practical work of the investigation and wrote the manuscript. R. A. Zaveri provided the MOSAIC code and assistance in its employment. All authors contributed to devising the methodology, result output and manuscript.

*Competing interests.* The authors declare that they have no conflict of interest.

5  *Acknowledgements.* The PhD of SO was funded by the Natural Environment Research Council award NE/K500859/1. This work was also funded by the Natural Environment Research Council award NE/M003531/1. Participation of RAZ in this study was supported by the US Department of Energy (DOE) Atmospheric System Research (ASR) Program under contract DE-AC06-76RLO 1830 at Pacific Northwest National Laboratory.



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
