# Peer review of "An efficient approach for treating composition-dependent diffusion within organic particles"

_Atmospheric Chemistry and Physics, 2016_

## Referee Comment (RC1) · Anonymous Referee #2 · 8 Feb 2017

The manuscript by O'Meara et al. describes a scheme that can efficiently treat the composition-dependent diffusion problem within aerosol particles. They used an improved numerical model as a benchmark, and developed a set of parameterizations for the correction of analytical solutions. This scheme was further implemented in the model MOSIAC to simulate the evolution of particle number size distribution. I think this is a timely paper as the non-liquid state and the associated slow in-particle diffusion has become a hot topic in current atmospheric chemistry research. This study was conducted carefully and the paper is informative. I would recommend publication in ACP once the authors address the specific comments below.

Specific comments

1. The authors mentioned water as an important plasticizer for organic particles. I think

this is also one of the motivations to develop such a composition-dependent diffusion model. However, it appears that the proposed correction scheme cannot be directly applied to the water/organic aerosol system as the molar volume ratio is far from the 1:1 ratio assumed in the simulations. This caveat should be discussed more explicitly in the manuscript.

2. It is not clear how the in-particle diffusion of the non-volatile component was considered in the model. It seems that Eq. 1 can be applied to both the semi-volatile and non-volatile components. If that is the case, does the value of D in Eq. 2 represent both D_sv and D_nv? This assumption is reasonable for the semi-volatile/non-volatile system discussed in this paper. These diffusivities, however, need to be treated separately if molecular sizes are vastly different (such as water/organics). Some more discussions would be helpful.

3. Page 2 Line 13: "Unlike gas-phase diffusion, which is already accounted for in regional-scale models by equations of gas/particle partitioning...". Equations of gas/particle partitioning not always account for gas-phase diffusion. Many models treat the gas/particle partitioning as equilibrium partitioning, i.e., gas-phase diffusion was not explicitly considered. However, it might be true that the timescale of gas phase diffusion is short enough compared to the typical time step in atmospheric models. Some models may use a dynamic gas/particle partitioning scheme where gas-particle mass transfer rates are taken into account. Please revise this sentence to be more specific.

4. Page 6 Line 3: The meaning of "ve $\Delta$xs,sv" is not clear. What does "ve" stand for? Some descriptions for "+ve $\Delta$xs,sv" and "-ve $\Delta$xs,sv" might be needed in the caption of figure 1, too (condensation and evaporation?).

5. Figure 3: It appears that the value of correction factor can be large even there is no composition dependence of D (pink and green lines in Fig. 3). This inaccuracy due to changing particle size could be further emphasized in the discussions and conclusions, too.

6. Figure 9a: instead of showing the discrepancy between the models with/without the correction, I think it would be better to show the difference for each model with respect to the numerical solution. This may help the readers to understand how the results of composition-dependent model are improved compared with the model with a constant D.

Technical comments:

Page 5 Line 14: "by a factor or e"->"by a factor of e"

---

## Referee Comment (RC2) · Anonymous Referee #1 · 23 Feb 2017

The authors have developed a correction factor for the analytical solution for particle phase diffusion to account for the composition dependence of the particle phase diffusion coefficient. The topic of the manuscript is timely and is well suited for ACP. The developed correction has weaknesses regarding the generality of the solution and this limits the applicability of the correction to wider range of the conditions. However, these limitations are described in the manuscript, and within the suitable range of conditions the correction works well. The methods used in the study seem in general valid and the manuscript can be published in ACP after addressing below comments.

Specific comments:

P. 1. L. 12-13: Authors state "Until such time as a general solution is found, caution should be given to sensitivity studies that assume constant diffusivity." I interpret this

as a critique towards using constant diffusivity approach. However, as the authors themselves admit that they were not able to derive a general composition dependent description, it would seem to me that constant diffusivity assumption is better choice at the moment than using a composition dependence that might not be suitable for the given conditions.

P. 5, L. 12-14: Authors chose e-folding state as the reference "time point". Did the authors test if the choice of this reference point affects the comparison results? Also, correct 'or' -> 'of' (L. 14)

P. 6, L. 7: It is stated that analytical solution was fit by eye to Fi-PaD results. Is fitting by eye enough accurate method for this? Why not using a more mathematical method for fitting the curve?

P. 6, L. 11-12: It says "An interpolation method was developed to estimate parameters for the correction equation between the values of dxs,sv and log10(D0nv/D0sv) used for the equation derivation". This developed interpolation method needs to be described. I assume this refers to Tables A3 and A4. However, these tables and their captions require clarification. For instance, please clarify what it means when it says "whether the log10 of parameter values was taken" and "whether the log10 of the variable was taken" in the table captions.

P. 7, L. 16-17: Why is the metric for proximity to equilibrium different for +ve and –ve cases?

P. 12, L. 3-9 and Figure 9: Is the % error presented in Fig. 9a comparing analytical constant D version to Fi-PaD results or to analytical composition dependent D version? Based on text and use of eq. 9 it seems that comparison point would be Fi-PaD, but based on figure caption it sounds like comparison point is the analytical solution with composition dependent D.

P. 14, eq. 12: Please explain why was this particular functional form chosen for the p4.

[Figure]

P. 15, L. 18-19: "In both solutions (numerical and analytical), diffusion rates have a square dependence on particle size". If one substitutes eq. 6 to eq. 3 the diffusion rate in eq. 3 is dependent on Rp not Rpˆ2. Did the authors test the CD for different sizes?

P. 15-16, from P. 15 L. 27 to P. 16 L. 3: If I understood correctly, the effect of different molar masses was tested by using the fitting parameter values that where determined from assuming both molar masses where 100 g/mol, and this CD did not work when M of semi-volatile was varying from that of non-volatile. Is it so that the correction factor CD simply doesn't work if M of compounds are different or would the CD work for the different M values if p parameters were fitted by using the M values that are of interest?

Technical notes:

P. 4, L. 24: Here it says subscript s refers to surface of particle, but eq. 3-5 do not contain subscript s. Is there a typo either in the equations or in the text?

P. 6, L. 3-4: Explain here what +ve and –ve mean.

P. 13, L. 13 and Fig. 10: I recommend naming the x-curves with some other symbol than p. Use of p here is confusing as also the fitting parameters for CD are marked with p.

---

## Author Comment (AC1) · 19 Apr 2017

Authors' Response to referee comments for 'An efficient approach for treating composition-dependent diffusion within organic particles'

Comments from referees are presented in blue, these are followed by author's response in green and changes to the manuscript are in red.

We thank both referees for their detailed and helpful comments. The resulting changes discussed below have improved the presentation, clarity and message of the paper.

Referee #1:

1. The authors mentioned water as an important plasticizer for organic particles. I think this is also one of the motivations to develop such a composition-dependent diffusion model. However, it appears that the proposed correction scheme cannot be directly applied to the water/organic aerosol system as the molar volume ratio is far from the 1:1 ratio assumed in the simulations. This caveat should be discussed more explicitly in the manuscript.

This is an important caveat given the relatively high abundance and high self-diffusion coefficient of water in the atmosphere. Consequently we have modified the text to express it with greater emphasis. In the discussion (page 16, line 19) we write:

The presented parameter values are therefore unable to reliably estimate diffusion when the molar volume ratio of components does not equal 1:1 (a value we chose arbitrarily). Parameter values account for changes to the diffusivity and diffusing distance due to partitioning of a semi-volatile component. The diffusing distance is dependent on component molar volume ratios, therefore when these are varied new parameters are required. For such systems, however, the presented parameterisation for the correction is valid. With different parameter values the parameterisation would be applicable to, for example, the case of diffusion in a mixture of water and organic material.

We note that our response to point 9 of referee #2 is relevant to this point.

2. It is not clear how the in-particle diffusion of the non-volatile component was considered in the model. It seems that Eq. 1 can be applied to both the semi-volatile and non-volatile components. If that is the case, does the value of D in Eq. 2 represent both $D_{sv}$ and $D_{nv}$? This assumption is reasonable for the semi-volatile/non-volatile system discussed in this paper. These diffusivities, however, need to be treated separately if molecular sizes are vastly different (such as water/organics). Some more discussions would be helpful.

The authors appreciate that some measurements show diffusion coefficients to be correlated with molecular size. However, in a two-component system where diffusion occurs along a mole fraction gradient, i.e., there is net movement of both components, the diffusion process is coupled. In order to attain volume continuity when using a fixed coordinate reference frame, the diffusion coefficients of the two components are then equal. This is explained in the paper with the addition of the following lines to the method (page 4, line 5):

The model described below uses a stationary coordinate reference frame, which for an ideal binary system requires that each component have the same diffusivity, quantified by the diffusion coefficient. This is true regardless of the component molecular size, and is physically necessary to attain volume continuity (Taylor and Krishna, 1993; Krishna and Wesselingh, 1997). The mathematical proof for the necessity of symmetric diffusion coefficients in an ideal binary system (e.g. comprising components 1 and 2) begins with:

$$\frac{-\nabla C_1}{\nabla C_2} = \frac{V_{m,2}}{V_{m,1}}$$

where $V_m$ is the molar volume and $\nabla C$ is the concentration gradient at the boundary where flux is being considered. Next, Fick's first law (which is equivalent to Fick's second law when flux is at steady state) can be expressed in terms of volumetric flux ($m^3 \, s^{-1}$):

$$J_i = D_i \nabla C_i V_{m,i} A$$

where A is the area diffusion occurs over. The magnitude of volumetric flux has to be equivalent for both components in order to attain volume continuity. For a particle of finite volume this means mass continuity is also satisfied. With this stipulation and eq. 3, it can be seen that symmetric diffusion coefficients are physically necessary.

3. Page 2 Line 13: "Unlike gas-phase diffusion, which is already accounted for in regional-scale models by equations of gas/particle partitioning...". Equations of gas/particle partitioning not always account for gas-phase diffusion. Many models treat the gas/particle partitioning as equilibrium partitioning, i.e., gas-phase diffusion was not explicitly considered. However, it might be true that the timescale of gas phase diffusion is short enough compared to the typical time step in atmospheric models. Some models may use a dynamic gas/particle partitioning scheme where gas-particle mass transfer rates are taken into account. Please revise this sentence to be more specific.

This sentence was considered to be unnecessary, and was removed with a slight modification of the following sentences to draw an appropriate link to regional-scale models (page 2, line 13):

To date particle-phase diffusion has not been included in regional-scale atmospheric models. Two outcomes of recent studies, however, indicate that particle-phase diffusion may pose a limitation to mass transfer over the timescales relevant to these models.

4. Page 6 Line 3: The meaning of "ve $\Delta x_{s,sv}$" is not clear. What does "ve" stand for? Some descriptions for "+ve $\Delta x_{s,sv}$" and "-ve $\Delta x_{s,sv}$" might be needed in the caption of figure 1, too (condensation and evaporation?).

The authors agree. The method has been changed to (page 6, line 4):

i) constant $x_{s,sv,eq}$ , with initial/equilibrium $x_{s,sv} = 0$ for positive (+ve, i.e. condensing case) $\Delta x_{s,sv}$ / negative (−ve, i.e. evaporating case) $\Delta x_{s,sv}$

and figure 1 caption has been changed to (page 6):

Figure 1. The shell resolution (given in the legend) distribution with $\Delta x_{s,sv}$ and $\log 10(D_{nv}^0 / D_{sv}^0)$ used, for: a) positive (+ve) $\Delta x_{s,sv}$ and b) negative (-ve) $\Delta x_{s,sv}$ .

5. Figure 3: It appears that the value of correction factor can be large even there is no composition dependence of D (pink and green lines in Fig. 3). This inaccuracy due to changing particle size could be further emphasized in the discussions and conclusions, too.

We agree, and have added the following lines to the results and discussion, respectively (page 8, line 15):

Where the self diffusion coefficients of components are the same in fig. 3, the correction is required only for the changing particle size.

& (page 15 line 10)

The correction was required to account not only for variable diffusivity but also varying particle size as the semi-volatile component partitions between phases, since the uncorrected analytical solution assumes constant particle size.

Inaccuracy due to particle size is also mentioned in the edit regarding point 1 of referee #1.

6. Figure 9a: instead of showing the discrepancy between the models with/without the correction, I think it would be better to show the difference for each model with respect to the numerical solution. This may help the readers to understand how the results of composition-dependent model are improved compared with the model with a constant D.

The suggested changes were made to fig. 9a (page 13):

[Figure]

**Figure 9.** In a), the discrepancy (found using eq. 10) in estimated radius with model run time normalised to the $e$-folding time ($t_e$) when $x_{s,sv}$ is increased instantaneously from 0.00 to 0.88 for two diffusion coefficient treatments: i) corrected analytical solution with $D_{nv}^0 = 1 \times 10^{-26}$ and $D_{sv}^0 = 1 \times 10^{-22}$ m$^2$s$^{-1}$ ($D_{sv,3}^0$) and ii) using the analytical without correction when $D$ is constant at $4.4 \times 10^{-23}$ m$^2$s$^{-1}$ ($D_{con}$). In later plots are the number size distributions for the same diffusion coefficient treatments, with red representing the former treatment (variable $D$) and blue the latter one (constant $D$)). In b) $t = 2.4 \times 10^4$ s, c) $t = 4.8 \times 10^4$ s and in d) $t = 1.2 \times 10^5$ s since simulation start.

Page 5 Line 14: "by a factor or e"->"by a factor of e"

Typographic error was corrected (page 5, line 24).

Referee #2

1. P. 1. L. 12-13: Authors state "Until such time as a general solution is found, caution should be given to sensitivity studies that assume constant diffusivity." I interpret this as a critique towards using constant diffusivity approach. However, as the authors themselves admit that they were not able to derive a general composition dependent description, it would seem to me that constant diffusivity assumption is better choice at the moment than using a composition dependence that might not be suitable for the given conditions.

This sentence in the abstract was presented without sufficient expansion in the main text. It is an important message because diffusivities measured at low relative humidities, for example, may not be applicable to the wider atmosphere where relative humidity and consequent water concentration in the particle-phase can change. We have added the following to the discussion so that a substitute to the constant diffusivity assumption is recommended (page 18, line 9):

In modelling studies where composition-dependent diffusion occurs and gas-phase saturation ratios of partitioning components vary over similar timescales to particle-phase diffusion, we recommend the numerical solutions mentioned above in preference to the assumption of constant diffusivity.

2. P. 5, L. 12-14: Authors chose e-folding state as the reference "time point". Did the authors test if the choice of this reference point affects the comparison results? Also, correct 'or' -> 'of' (L. 14)

Following this comment, models were run until the difference between surface and bulk concentration of the semi-volatile decreased by a factor of $16e$. Although good agreement between the numerical and corrected analytical solutions was seen for the evaporating case this was not true for the condensation case. Consequently the correction equation used for the evaporation case was used and fitting was repeated, giving the new parameter values shown in table A3. These do give good agreement to this extended equilibrium point. Several changes resulted in the manuscript:

Figs. 3, 4, 5, 7, 8, 9 and 11 were reproduced with the new correction implemented.

New parameters in Table A1 and new interpolation method in Table A3.

Page 8, line 9:

The correction factor was then plotted against proximity to equilibrium; here we use the absolute difference between surface and bulk average $x_{sv}$. This process was done for the model inputs shown in fig. 1 to determine whether a general equation form could be found that described the relationship between the D correction factor ($C_D$) and proximity to equilibrium. Examples are shown in fig. 3. The resulting general equation for both +ve and -ve $\Delta x_{s,sv}$ is found to be:

$$C_D = e^{((|x_{s,sv}-x_{a,sv}|)^{p1})p2} - p3,$$

where $p_n$ is a parameter value, dependent on $\Delta x$ and $D^0_{nv}/D^0_{sv}$.

The testing to an extended equilibrium point is now stated (page 8, line 29):

Corrected analytical and numerical solution results were also compared beyond the e-fold time, until the difference in concentration between surface and bulk had diminished to a factor of 16e. The agreement shown in fig. 4 was maintained to this further equilibration point.

3. P. 6, L. 7: It is stated that analytical solution was fit by eye to Fi-PaD results. Is fitting by eye enough accurate method for this? Why not using a more mathematical method for fitting the curve?

A more mathematical method was attempted through least squares fitting, however, it was found to be under constrained. We consider the assessment of fit through comparison of radius measurements to provide an objective means of assessing the accuracy of the correction. We have added the following text to the method to

explain this (page 6, line 9):

To derive correction equations $R_p - t$ profiles estimated by the analytical solution were fit to those of Fi-PaD. A least squares fitting procedure was attempted and found to be under constrained, thus fitting was done by eye, and the quality of fit was objectively assessed through residuals, as described below.

4. P. 6, L. 11-12: It says "An interpolation method was developed to estimate parameters for the correction equation between the values of dxs,sv and log10(D0nv/D0sv) used for the equation derivation". This developed interpolation method needs to be described. I assume this refers to Tables A3 and A4. However, these tables and their captions require clarification. For instance, please clarify what it means when it says "whether the log10 of parameter values was taken" and "whether the log10 of the variable was taken" in the table captions.

The authors agree that greater explanation and clarity was needed for explaining the interpolation procedure. The following was added to the results (page 9, line 5):

The general method was to first interpolate with respect to $\Delta x_{s,sv}$ followed by $D_{nv}^0 \big/ D_{sv}^0$. For most accurate results the interpolation equation was found to be dependent on the independent variables as described in the appendix (Tables A3 and A4). The interpolation was tested at $\Delta x_{s,sv}$ and $D_{nv}^0 \big/ D_{sv}^0$ comparatively far from those with known parameter values and spread across the variable space.

and the table A3 caption was changed to (page 22):

Interpolation method for parameters in eq. 9 (for +ve $\Delta x_{s,sv}$). Interpolation is done with respect to $\Delta x_{s,sv}$ first, followed by $\log_{10}(D_{nv}^0 \big/ D_{sv}^0)$; the method for the former is given in the upper part of the table and the method for the latter is in the lower part. Note the method is dependent on the independent variable. Methods are represented by codes. The first number in each code is 1 if interpolation is done with respect to the $\log_{10}$ of parameter values, in which case the interpolation result must be raised to the power 10, and is 0 if no logarithm is taken. The second number in each code is 1 if the interpolation is done with respect to the $\log_{10}$ of the independent variable, and is 0 if no logarithm is taken. The final letter represents the form of the interpolation: L and S for linear and spline, respectively. For p2, when interpolating with respect to $\log_{10}(D_{nv}^0 \big/ D_{sv}^0)$, the interpolation method depends on the value of this variable, which is denoted $D^0$.

and the table A4 caption was changed to (pag 22):

Interpolation method for parameters in eq. 9 (for -ve $\Delta x_{s,sv}$). Interpolation is done

with respect to $\Delta x_{s,sv}$ first, followed by $\log_{10}(D^0_{nv}/D^0_{sv})$; the method for the former is given in the upper part of the table and the method for the latter is in the lower part. Note the method is dependent on the independent variable. Methods are represented by codes. The first number in each code is 1 if interpolation is done with respect to the $\log_{10}$ of parameter values, in which case the interpolation result must be raised to the power 10, and is 0 if no logarithm is taken. Because parameters are sometimes negative, to gain a real result from the logarithm, a constant must be added to the parameters first, if this is the case this constant is given in brackets beside the first code number (once interpolation is complete this constant is subtracted from the result after it has been raised to the power 10). The second number in each code is 1 if the interpolation is done with respect to the $\log_{10}$ of the independent variable, and is 0 if no logarithm is taken. The final letter represents the form of the interpolation: L and S for linear and spline, respectively. For p2, when interpolating with respect to $\log_{10}(D^0_{nv}/D^0_{sv})$, the interpolation method depends on the value of this variable, which is denoted $D^0_{rat}$.

5. P. 7, L. 16-17: Why is the metric for proximity to equilibrium different for +ve and –ve cases?

Please see our response to referee #2, point 2

6. P. 12, L. 3-9 and Figure 9: Is the % error presented in Fig. 9a comparing analytical constant D version to Fi-PaD results or to analytical composition dependent D version? Based on text and use of eq. 9 it seems that comparison point would be Fi-PaD, but based on figure caption it sounds like comparison point is the analytical solution with composition dependent D.

Please see our response to referee #1, point 6.

7. P. 14, eq. 12: Please explain why was this particular functional form chosen for the p4.

There was insufficient explanation in the text of why a new equation was necessary for the case of varying surface mole fraction of the semi-volatile component. We have added the following to provide an explanation (page 14, line 11):

A new correction equation was required that could accommodate a variable surface mole fraction and give agreement with Fi-PaD estimates. Through fitting by eye this was found to be:

8. P. 15, L. 18-19: "In both solutions (numerical and analytical), diffusion rates have a square dependence on particle size". If one substitutes eq. 6 to eq. 3 the diffusion rate in eq. 3 is dependent on Rp not Rp^2. Did the authors test the CD for different sizes?

This comment highlighted an omission in the original manuscript - the units for concentration in the particle-phase bulk for eq. 5 were not stated. They are mol m^-3 (air) rather than mol m^-3 (particle). This, and the result in terms of dependence of diffusion rate on particle size is stated in a modified paragraph following eq. 7 (previously eq. 5) (page 5, line 5):

where $a$ represents the bulk of the particle-phase, $g$ represents the gas-phase, j is the index for all components, m is the index for size-bin, $R_p$ is particle radius, $C_g^*$ is the effective saturation vapour concentration ($mol m^{-3}$(air)), C is the concentration in the bulk part of a phase ($mol m^{-3}$(air)) and N is the particle number concentration ($m^{-3}$(air)). In order to compare results from eq. 1 and eq. 5, concentrations from the latter must be divided by the volume concentration of particles ($m^3$(particle)$m^{-3}$(air)). Following this division, it can be seen that diffusion has an inverse square dependence on particle radius in both solutions.

9. P. 15-16, from P. 15 L. 27 to P. 16 L. 3: If I understood correctly, the effect of different molar masses was tested by using the fitting parameter values that where determined from assuming both molar masses where 100 g/mol, and this CD did not work when M of semi-volatile was varying from that of non-volatile. Is it so that the correction factor CD simply doesn't work if M of compounds are different or would the CD work for the different M values if p parameters were fitted by using the M values that are of interest?

The CD would work for systems of different molar volume ratios than used here. We agree that this was not clearly stated. We have expressed this point better by modifying the discussion. We now say (page 16, line 19):

The presented parameter values are therefore unable to reliably estimate diffusion when the molar volume ratio of components does not equal 1:1 (a value we chose arbitrarily). Parameter values account for changes to the diffusivity and diffusing distance due to partitioning of a semi-volatile component. The diffusing distance is dependent on component molar volume ratios, therefore when these are varied new parameters are required. For such systems, however, the presented parameterisation for the correction is valid. With different parameter values the parameterisation would be applicable to, for example, the case of diffusion in a mixture of water and organic material.

We note that the same modification was used to address point 1 of referee #1.

Furthermore, line 18 of page 17 was changed to:

Along with the effect of component molar volume ratios on diffusion, however, this could be overcome through refitting of parameter values.

and line 20 of page 18 was changed to:

The verified correction is currently limited to conditions of similar molar volume between the partitioning component and the particle average, and of a logarithmic dependence of diffusion coefficient on partitioning component mole fraction. These limitations may be overcome through refitting of parameters.

10. P. 4, L. 24: Here it says subscript s refers to surface of particle, but eq. 3-5 do not contain subscript s. Is there a typo either in the equations or in the text?

This typo was fixed (now page 5, line 5).

11. P. 6, L. 3-4: Explain here what +ve and –ve mean.

This was done, as described in our response to point 4 of referee #1.

12. P. 13, L. 13 and Fig. 10: I recommend naming the x-curves with some other symbol than p. Use of p here is confusing as also the fitting parameters for CD are marked with p.

This was done, with p changed to prof (pages 14 and 15).

---

## Author Response (AR2)

**Authors' Response to second round of referee comments for 'An efficient approach for treating composition-dependent diffusion within organic particles'**

Comments from referees are presented in blue, these are followed by author's response in green and changes to the manuscript are in red.

Referee #1:

1. In the revised manuscript the authors have addressed my comments appropriately. However, I find that one point still requires clarification in the manuscript. Therefore, I recommend that the manuscript is published after the below comment has been addressed.

In my original comment I was questioning the fitting of the parameters by eye. In their reply the authors state that "A more mathematical method was attempted through least squares fitting, however, it was found to be under constrained." If I understood correctly, the authors first fitted the R-t profiles by eye to get the parameters and then goodness of the final fit was evaluated with the Eq. (10). The fact that the values were found by fitting by eye should not be hidden from a reader. This information is important if a reader wishes to use the given parameterization, and, therefore, requires to know how the values were determined. So I strongly recommend adding 'fitting by eye' on the page 8 line 21 from where it has been removed during the revision of the manuscript.

The authors agree this should be made clear, so have changed page 8 line 21 to:

[revised manuscript text omitted]